# Proteomic Characterization of Virulence Factors and Related Proteins in *Enterococcus* Strains from Dairy and Fermented Food Products

**DOI:** 10.3390/ijms231810971

**Published:** 2022-09-19

**Authors:** Ana G. Abril, Marcos Quintela-Baluja, Tomás G. Villa, Pilar Calo-Mata, Jorge Barros-Velázquez, Mónica Carrera

**Affiliations:** 1Department of Microbiology and Parasitology, Faculty of Pharmacy, University of Santiago de Compostela, 15898 Santiago de Compostela, Spain; 2Department of Food Technology, Spanish National Research Council (CSIC), Marine Research Institute (IIM), 36208 Vigo, Spain; 3Department of Analytical Chemistry, Nutrition and Food Science, Food Technology Division, School of Veterinary Sciences, University of Santiago de Compostela, Campus Lugo, 27002 Lugo, Spain

**Keywords:** LC–ESI–MS/MS, proteomics, mass spectrometry, antibiotic resistance peptides, antibiotic production, virulence factors, *Enterococcus* spp., dairy, fermented food products

## Abstract

*Enterococcus* species are Gram-positive bacteria that are normal gastrointestinal tract inhabitants that play a beneficial role in the dairy and meat industry. However, *Enterococcus* species are also the causative agents of health care-associated infections that can be found in dairy and fermented food products. Enterococcal infections are led by strains of *Enterococcus* *faecalis* and *Enterococcus* *faecium*, which are often resistant to antibiotics and biofilm formation. Enterococci virulence factors attach to host cells and are also involved in immune evasion. LC-MS/MS-based methods offer several advantages compared with other approaches because one can directly identify microbial peptides without the necessity of inferring conclusions based on other approaches such as genomics tools. The present study describes the use of liquid chromatography–electrospray ionization tandem mass spectrometry (LC–ESI–MS/MS) to perform a global shotgun proteomics characterization for opportunistic pathogenic *Enterococcus* from different dairy and fermented food products. This method allowed the identification of a total of 1403 nonredundant peptides, representing 1327 proteins. Furthermore, 310 of those peptides corresponded to proteins playing a direct role as virulence factors for *Enterococcus* pathogenicity. Virulence factors, antibiotic sensitivity, and proper identification of the enterococcal strain are required to propose an effective therapy. Data are available via ProteomeXchange with identifier PXD036435. Label-free quantification (LFQ) demonstrated that the majority of the high-abundance proteins corresponded to *E. faecalis* species. Therefore, the global proteomic repository obtained here can be the basis for further research into pathogenic *Enterococcus* species, thus facilitating the development of novel therapeutics.

## 1. Introduction

*Enterococcus* species include a ubiquitous group of Gram-positive organisms present in natural environments, playing a beneficial role in the dairy and meat industry for dairy and fermented food products, and in the gastrointestinal (GI) tract of humans and other animals. These bacteria are also one of the major causative agents of health care-associated infections. Enterococcal infections are often resistant to antibiotics and biofilm production [1,2].

*Enterococcus faecalis* and *Enterococcus faecium* cause the majority of enterococcal infections (urinary tract, catheterized urinary tract, bloodstream, wounds and surgical sites, and heart valves in endocarditis), while *E. faecalis* is responsible for 80–90% of enterococcal-associated nosocomial infections, followed by *E. faecium* (10–15%). As common inhabitants of the human GI tract, these organisms may be considered opportunistic pathogens and are involved in polymicrobial diseases [3,4].

Thus, the pathogenesis of Enterococci may be achieved by their production of virulence factors, in addition to their resistance to antimicrobials. Virulence factors play a role in attachment to host cells or extracellular matrix (ECM) proteins and are also involved in immune evasion [5]. Specifically, the production of cell surface proteins, the capacity for maintaining the cell envelope integrity, the adaptation to whatever nutrient sources are available, and potentially the formation of biofilms readily promote GI colonization and have important projections of their ability to participate in HGT processes [6]. Moreover, the presence of chromosomally integrated bacteriophages and plasmids contributes to the plasticity of enterococcal genomes and thus the evolution of multidrug-resistant enterococci [7]. In addition, to promote their competitive fitness with bacterial commensals in the GI tract, they may produce antimicrobials often genetically encoded on conjugative plasmids [6].

*Enterococcus* detection and identification have usually focused on culture and biochemical tests [8,9]. The methodology involved is time-consuming and inappropriate for short shelf-life foodstuffs that require fast and unequivocal methods to quickly detect and identify pathogens. Although several procedures that include molecular-based techniques, such as polymerase chain reaction (PCR), whole genome sequencing, and enzyme-linked immunosorbent assay (ELISA), all of them requiring long enrichment times, expensive chemicals and specialized equipment, have been developed [10,11,12], other methods, including biosensors, can be applied [13]. Furthermore, several studies that dig deeper into the *Enterococcus* proteome analysis have been proposed as alternative methods of fast enterococcal identification, as occurs with the ability to form biofilms [14,15]. In addition, matrix-assisted laser desorption/ionization time-of-flight mass spectrometry (MALDI-TOF MS) [16,17] was used to establish antimicrobial resistance classification among the *Enterococcus* spp. Another approach has been performed by liquid chromatography coupled to tandem mass spectrometry (LC–MS/MS) to analyze the proteome of a dairy-isolated *E. faecalis* [18].

The detection of antimicrobial resistance mechanisms and virulence factors in enterococci has been traditionally performed using molecular genetic methods. However, while this information provides the repertoire of genetic mechanisms present in the bacterial genome, further proteomic analyses could offer information on the proteins produced in a given environment and proteomic fingerprinting for microbial identification. Techniques such as liquid chromatography–electrospray ionization tandem mass spectrometry (LC-ESI-MS/MS) have been used successfully to specifically identify pathogenic bacterial strains [19,20]. Additional studies involve the use of LC–ESI–MS/MS to research the proteomics of antibiotic resistance and the production of antimicrobials and other virulence factors to correctly identify different streptococcal and *Listeria* species [21,22].

This study aims to identify bacterial peptides related to virulence factors, antimicrobials, antimicrobial resistance mechanisms, and toxins using advanced shotgun proteomic methods (LC-ESI-MS/MS). This proteomic characterization of dairy-related enterococci would be important for detecting disease outbreaks quickly, providing a broad context for epidemiological investigations from a One Health approach.

## 2. Results

### 2.1. Enterococcus *spp.* Proteomics Data Repository

Fourteen *Enterococcus* strains were studied (Table 1). Bacterial peptides were prepared by treating the protein mixtures with trypsin and analyzed by LC-ESI-MS/MS, as described previously [19,21,22,23,24,25]. A total of 1403 nonredundant peptides were identified, which corresponded to 1327 annotated proteins from the *Enterococcus* UniProt/TrEMBL database (431,881 protein sequence entries in September 2020). The mass spectrometry proteomics data have been deposited to the ProteomeXchange Consortium via the PRIDE [26] partner repository with the dataset identifier PXD036435. The depth of the proteome results corresponded to surface-associated proteins due to the fact that the majority of virulence factors are localized in the surface of bacteria, either secretory or membrane associated [27,28], and we prepared the protein extraction to increase its identification. Accordingly, the virulence factors were identified by comparison with both the “Virulence Factors of Pathogenic Bacteria Database” (VFDB; http://www.mgc.ac.cn/VFs/, accessed on 20 May 2022) and with previously reported data [1,6,29,30,31].

In the group of nonredundant peptides, 310 were unequivocally identified as virulence factors and included proteins such bacteriocins, multidrug transporters, and phage-associated proteins. Additional polypeptides (toxins and antitoxins), together with polypeptides involved in antibiotic resistance, were also found. The 310 virulent factors identified for the studied strains are displayed in Appendix A and organized into groups according to the main role they play; these virulent factors include toxins, antibiotic resistance peptides, and other tolerance proteins involved in resistance to toxic substances, colonization and immune evasion factors, antimicrobial compounds, ATP-binding cassette (ABC), and other transporters associated with virulence factors. In addition, the main antibiotic resistance proteins, antimicrobial-related proteins, and other virulence factors are summarized in Table 2. Table 3 displays the total number of peptides corresponding to virulence factors organized into groups according to their main role.

Strains F5, F2, F1, F3, and F13 contained the highest number of peptides related to virulence, with 50, 44, 39, 37, and 32 peptides of virulence, respectively. Taken together, these results suggest that these strains are probably the most pathogenic strains within the species *E. faecalis*. However, for strains F4, F10, and F12, fewer than 50 nonredundant total peptides were identified (see the complete nonredundant dataset in Excel Appendix A).

### 2.2. Label-Free Quantification (LFQ) of Enterococcus Species

Relative label-free quantification of each Enterococcus species (*E. faecalis*, *E. faecium* and *E. durans*) was also performed to determine the protein abundance of each sample. Appendix A contains these results. High-abundance proteins for each species and strains were compared. Figure 1 shows the distribution of the high-abundance proteins detected for each strain.

According to the different species, Figure 2 shows the high-abundance proteins for each Enterococcus species (*E. durans*, *E. faecalis,* and *E. faecium*).

Using a Venn diagram, Figure 3 shows the distribution and overlapping of the high-abundance proteins for all the *Enterococcus* samples analyzed by LFQ.

As was demonstrated in Figure 1, Figure 2 and Figure 3, the majority of the high-abundance proteins were detected in *E. faecalis* species.

### 2.3. Proteins Involved in Bacterial Resistance to Antibiotics and Other Toxic Substances

This study identified 32 enterococcal peptides related to antimicrobial resistance or toxic substances (Appendix A). Twenty-one of the proteins determined were associated with antibiotic resistance, whereas the remaining eleven peptides were related to other tolerances.

Two peptides were characterized as belonging to the MarR family of transcriptional regulators, and a set of three peptides was characterized as penicillin-binding proteins, which exhibit high affinity for the antibiotic [32]. MarR acts as a regulator for proteins involved in resistance against several antibiotics [33]. Three peptides belonging to the TetR family of regulators (TFRs) were identified; TetR proteins play a role in regulating genes encoding small-molecule exporters and antibiotic resistance. They also contribute to both antibiotic and quorum-sensing production [34]. Another peptide has been identified as a PASTA domain (penicillin-binding protein and serine/threonine kinase associated domain), which is found mainly in Gram-positive bacteria [35,36]. A transmembrane Ser/Thr kinase (IreK) in the PASTA kinase protein family is involved in cell envelope integrity and antimicrobial resistance in *E. faecalis* [37].

Three additional peptides were found to correspond to the GCN-2-related N-acetyl transferase (GNAT) family of acetyltransferases, which confer antibiotic resistance by catalyzing the acetylation of amino groups in aminoglycoside antibiotics [38]. VanY D-Ala-D-Ala carboxypeptidase was also identified by analysis of their peptides, and vanY is necessary for the synthesis of the vancomycin-inducible D,D-carboxypeptidase. VanY D-Ala-D-Ala peptidases provide resistance to the antibiotic vancomycin in some strains of enterococci [39,40]. One peptide interestingly related to a DrrC protein, which is involved in resistance to daunorubicin, was also identified [41]. Moreover, a peptide has been found to belong to the OmpR/PhoB subfamily of proteins. Members of the OmpR/PhoB subfamily include diverse transcriptional regulators, such as *Enterococcus faecium* VanR, which controls resistance to the antibiotic vancomycin [42]. One of the peptides involved in the stress caused by phenolic acids was also identified, PadR, an environmental sensor that acts as a repressor of padA gene expression in the phenolic acid stress response [43]. An additional peptide was characterized as aminoglycoside N(3)-acetyltransferase [44]. This enzyme can provide resistance against a variety of antibiotics, including gentamicin, kanamycin, tobramycin, neomycin, and apramycin, which contain 2-deoxystreptamine rings; these rings act as acceptors for acetyltransferase activity.

Bacterial tolerance refers to the ability of bacterial populations to survive harsh environments, such as the environment created by the use of antimicrobial agents, without developing resistance, since many of the mechanisms governing bacterial tolerance directly influence the virulence of the strain. This study revealed peptides with a role in bacterial resistance and tolerance to such conditions. One peptide was identified as TelA, a protein belonging to the toxic anionic resistance family, which confers tellurite resistance [45]. Another further peptide corresponded to a Cass 2 protein. Cass 2 is an integron-associated protein that binds cationic drug compounds with submicromolar affinity [46]. Two peptides were related to mercury resistance MerR proteins; these peptides are metal ion sensing regulators that create a mercury-resistant phenotype via transcription of several Mer genes [47]. A peptide of a SugE protein was also determined; SugE is a small multidrug resistance protein [48].

Additional peptides related to other important bacterial tolerances, including thermotolerance and osmotolerance, were identified. Regardless, stress tolerances play a role in facilitating bacterial persistence in the environment. Accordingly, this study identified two peptides belonging to heat shock proteins, such as DnaJ, a chaperone involved in thermotolerance phenotypes. Another peptide was identified as belonging to the cold shock-like proteins (CSPs) that provide tolerances in extreme temperatures by cellular physiology modifications, including decreased membrane fluidity, reduced mRNA transcription and translation due to the stabilization of secondary structures, inefficient folding of some proteins, and reduced enzyme activity [49]. Two peptides, CspD and CspA, were identified. In addition, three peptides of bacterial general stress response protein, one of them characterized as CsbD [50] and another one as YitT, are involved in general stress protein and are particularly required for protection against paraquat stress [51].

Many bacterial transporters, such as the ABC transporter, play a role in either antibiotic and other resistances or tolerances. Some transporter proteins are described in Section 2.6.

### 2.4. Antibacterial Compounds and Proteins Involved in Antibacterial Production

This study identified six antibacterial proteins from all the bacterial strains analyzed, which are present in the proteomic repository for *Enterococcus* spp. (Appendix A). Bacteriocin-related peptides were discovered in the F1, F2, F5, F11, and F13 strains. Four peptides were identified with homology to lantibiotic biosynthesis proteins such as LanM, which is involved in posttranslational modifications [52]. Furthermore, one bacteriocin peptide was identified. The last bacteriocin-related peptide belongs to the radical S-adenosylmethionine (SAM enzymes, which tend to be involved in the maturation of subtilisin, anaerobic sulfatase-maturing enzyme, pyrroloquinoline quinolone (PQQ), and mycofactocin) [53].

### 2.5. Proteins Involved in Bacterial Toxicity

The present study identified nine peptides involved in bacterial toxicity by LC-ESI-MS/MS (Appendix A). The peptides correspond to PIN toxin domains; Type II toxin–antitoxin systems, such as Phd repressor/antitoxin, PemK/MazF family toxin, and RelE/ParE family toxin; and LXG domain-containing protein, AbrB family toxin, and exfoliative toxin A/B. Exfoliative toxins (ETs) are serine proteases that hydrolyze desmoglein 1 (Dsg1), which causes dissociation of keratinocytes in both human and animal skin, thus helping *S. aureus* colonize the skin of mammals [54]. AbrB family members are transcription factors that act as antitoxins [55]. RelE/ParE family toxin is a member of the Type II toxin–antitoxin system that was also identified in this study. RelE toxins are mRNA interferases, while ParE toxins inhibit gyrase activity [56]. One of the characterized peptides corresponds to a Type II toxin–antitoxin system PemK/MazF family toxin; this family of proteins contains a toxin and an antitoxin gene pair as part of a postsegregation killing system, where gene loss results in the toxin attacking the cell. MazE is the antitoxin that inhibits toxin MazF, and under stress conditions, mazEF transcription is reduced, leading to the degradation of MazE, thus inhibiting cell division and resulting in cell death [57]. Two peptides that belong to an LXG domain-containing protein were identified in the *Enterococcus* strains (F5 and F13); this domain is present in the N-terminal region of a group of polymorphic toxin proteins. In prokaryotes, PIN domains are the toxic components of toxin–antitoxin (TA) systems, and their toxicity is produced by their ribonuclease activity. The PIN domain TA systems are now called VapBC Tas (virulence associated proteins), where VapB is the inhibitor and VapC is the PIN-domain ribonuclease toxin [58].

### 2.6. Proteins Involved in Host Colonization and Immune Evasion

A total of 104 peptides were identified that belonged to proteins that play a role in *Enterococcus* colonization and immune evasion (Appendix A).

Enterococcal cell surface proteins are important for bacterial internalization into the mammalian host, playing a role in the maintenance of cell envelope integrity, adaptation to whatever nutrients are available, and formation of biofilms. To date, this study identified one peptide corresponding to a homolog of various *Listeria monocytogenes* virulence factors called internalin, a cell adhesion protein involved in host colonization [59]. Bacterial surface proteins are a group with important functions, such as adherence, invasion, signaling and interaction with the host immune system or environment in general. In Gram-positive bacteria, many surface proteins belonging to the “LPxTG” family are anchored to the peptidoglycan (PG) by an enzyme known as sortase [60]. Other adhesion peptides are indeed homologs to proteins involved in cell adhesion, such as thirteen peptides that corresponded to LPXTG-domain-containing protein, cell wall anchor domain, and one peptide assimilable to a sortase. LPXTG surface adhesin is involved in *E. faecium* biofilm formation [61]. Sortases are polypeptides that covalently attach secreted proteins to their cell wall to assemble pili; they play a key role during the infection process and represent potential drug targets [62]. Sortase A (SrtA) has been described as a membrane-associated enzyme that anchors surface proteins to enterococcal cell wall components, thus promoting biofilm formation [63,64].

Four additional peptides were found to be homologs to BspA, one peptide to Fibronectin/fibrinogen-binding protein, and one peptide to a collagen-binding protein. BspA is an antigen I/II family polypeptide that confers adhesion in Group B *Streptococcus* and is therefore linked to pathogenesis [65]. Adhesion pili are virulence factors present on the surface of bacteria; they are usually required for biofilm formation, enabling the bacteria to bind and adhere to target cells [66]. Four peptides were determined to have a fimbrial isopeptide-formation D2 domain. Proteins with this domain include fimbrial proteins with lectin-like adhesion functions, and most characterized members are involved in surface adhesion to host structure [67]. A peptide of flagellar hook-associated protein 2 (HAP2 or FliD) and an Fn3-like domain-containing protein peptide have also been identified. HAP2 forms the distal end of the flagella and plays a role in mucin-specific bacterial adhesion [68]. Fn3 is a fibronectin type III glycoprotein of the extracellular matrix that binds to membrane-spanning receptor proteins called integrins [69]. Finally, an uncharacterized adhesion protein was identified together with DUF4097, a domain-containing protein and an Ig domain-containing protein. Proteins that contain an Ig domain are found in a variety of bacterial and phage surface proteins, such as intimins, as well as in some uncharacterized eukaryote proteins. Intimin is a bacterial cell-adhesion molecule that mediates the intimate interaction between bacteria and host cells [70,71]. However, the DUF4097 domain-containing protein has a putative all-beta structure with a twenty-residue repeat with a highly conserved repeating GD, gly-asp, motif, and it may form part of a bacterial adhesion [72]. A pilin subunit SpaA was also identified, which is composed of the shaft pilin SpaA, the tip adhesion SpaC, and the base pilin SpaB anchored to the cell wall.

A total of 33 peptides were identified as peptidases and proteases, including members of the P60, M20, M74, M24, dipeptidase PepV and T peptidase families, CLp proteases, and oligoendopeptidase PepF/M3 family protein. In particular, the 60-kDa extracellular protein (p60) is a member of the P60 family that is encoded by the *iap* gene and participates in the host invasion process [73]. Moreover, the dipeptidase PepV (an enzyme located in the final stage of the intracellular proteolytic system) has been demonstrated to be distributed widely in lactic acid bacteria, especially in lactococci [74]. Additionally, a peptide that belongs to an oligoendopeptidase pepF/M3 family has been determined, and it can participate in the regulation of sporulation [75] (Kanamaru et al., 2002). ImmA/IrrE proteases, which are involved in bacterial resistance to hostile environments, have been identified in the analyzed strains [76] (Gómez et al., 2020). Moreover, an immune inhibitor A peptide has been identified; this protein belongs to the MEROPS peptidase family M6 (immune inhibitor A family). *B. thuringiensis* has two proteins belonging to this group (InhA and InhA2), and InhA2 has been shown to have a vital role in virulence when the host is orally infected. The *B. cereus* member has been found to be an exosporium component from endospores [77].

Moreover, two peptides belonging to the CLp ATase family were detected (CLpA and CLpC); CLp proteins are formed by a CLp ATase and a peptidase; the latter hydrolyzes the proteins controlling the modulation of virulence factors, such as biofilm formation [78,79].

This study characterized two additional peptides related to capsular polysaccharides (CPSs). CPS contributes to pathogenesis by inhibiting the entrapment of pneumococci in neutrophil extracellular traps; the cpsABCD locus is involved in the modulation of CPS biosynthesis [80,81]. In addition, one N-acetylmuramoyl-L-alanine amidase peptide was identified. Cbp proteins include an N-acetylmuramoyl-L-alanine amidase in their biological module, which is involved in peptidoglycan release, proinflammatory teichoic acid formation, cell division and bacterial colonization [82]. N-Acetylmuramoyl-L-alanine amidases are autolysins involved in bacterial adherence to eukaryotic cells. Additional lysis proteins identified three listerial peptides; the one contained an autolysin modifier protein, and two had peptides belonged to the LysM domain; this last one is a protein module, originally found in enzymes that degrade bacterial cell walls, present in many bacterial proteins thus far involved in pathogenesis [83]. Additionally, a peptide of a hemolysin III family channel protein was identified; this protein is an integral membrane protein with hemolytic activity and is found in different *Enterococcus* species [84].

An additional group of eleven peptides corresponded to an Mga protein, a DNA-binding protein that regulates the expression of virulence genes; MafR is a newly described member of the Mga/AtxA family of global transcriptional regulators found in *Enterococcus* [85]. The peptides found included the M protein family of polypeptides (emm, mrp, and enn), C5a peptidase (scpA), and collagen-like protein 1 (scl1), which play an important role in colonization and immune evasion [86]. Three of the peptides identified belonged to the well-characterized type VII secretion systems (ESSs). This system facilitates the secretion of extracellular proteins across the cytoplasmic membrane and is involved in host infection, which is associated with virulence in *S. aureus* [87].

Transcriptional regulators controlling virulence factors were also identified in *Enterococcus* species; they include one peptide identified as LysR, two peptides corresponding to LytR, and one peptide representing the LytTR transcriptional regulators. LysR contributes to virulence by controlling multiple pathways that include cationic antimicrobial peptide (CAMP) resistance, fructose and mannose metabolism, and beta-lactam resistance [88,89]. LytR and LytTR proteins regulate additional virulence factors, such as extracellular polysaccharides, toxins, and bacteriocins, and polypeptides bind specific DNA sequences and act as transcriptional activators [90,91,92].

One peptide belonging to the kdgR transcriptional regulator was also detected. kdgR is a LysR family regulator, a repressor linked to the synthesis of enzymes involved in pectin degradation, as well as pectinase(s) secretion and catabolism thereof, but mostly overall affecting the production of cell wall-degrading enzymes. These are secreted mainly through the Type II secretion system (T2SS) and are directed toward the breakdown of the host plant cell wall [93]. In addition, a peptide was identified as belonging to the ArpU family transcriptional regulator, which represents a group of proteins that includes the putative autolysin regulatory protein ArpU. ArpU was originally described as a regulator of cellular muramidase-2 of *Enterococcus hirae* but appears to have been cloned from a prophage [94].

One peptide belonging to the toxin secretion/phage lysis holin was also detected. Toxin secretion/phage lysis, holin, includes, in addition to phage holins, the protein TcdE/UtxA, a holin-like protein encoded by toxigenic isolates of *Clostridioides difficile* and related species. TcdE mediates the release of the large clostridial glucosylating toxins (LCGTs) that act in combination with lytic enzymes in bacterial lysis (Tan et al., 2001; Vidor et al., 2022).

A peptide obtained from the analyzed *Enterococcus* strains was identified as ComEA, and two others were identified as ComK; they are part of the Com system, which is involved in escaping from the host phagosome and reaching the cytoplasm. This role is facilitated by the competence (Com) system proteins [95]. The ComK regulator includes YlbF and YmcA proteins, which are required for competence development, sporulation and the formation of biofilms. Furthermore, additional peptides included methyl-/ethyl-accepting chemotaxis. These proteins are a family of bacterial receptors that mediate chemotaxis to diverse signals, responding to changes in the concentration of attractants and repellents in the environment, which results in the alteration of swimming behavior [96].

Two peptides were determined to be sporulation-related proteins; one peptide was a spore coat protein, and the other was the sporulation TjcZ protein, which is involved in spore germination. Proteins in this entry are found only in endospore-forming bacterial species. A Gly-rich variable region is followed by a strongly conserved, highly hydrophobic region, predicted to form a transmembrane helix, ending with an invariant Gly [97].

Three peptides of restriction enzymes have been determined, a type-2 restriction enzyme, a Sfel restriction endonuclease, and a Mrr, a type IV restriction endonuclease, both involved in the acceptance of modified foreign DNA and with the ability to restrict both adenine- and cytosine-methylated DNA. It constitutes an essential mechanism for the generation of genetic variability that in turn mediates adaptations to the environment in bacterial populations. Mrr spurs the SOS response after high-pressure stress in *Escherichia coli* [98].

Enterococcal polysaccharide antigen (EPA) is required for normal cell growth and division and for resistance to cephalosporins, playing a critical role in host colonization. EPA residues consist of phosphopolysaccharide chains corresponding to teichoic acids [99]. In addition, lipoteichoic acid (LTA) has been reported to be involved in a wide range of inflammatory diseases; LTA is recognized by eukaryotic Toll-like receptor 2 (TLR2), which triggers innate immune responses [100]. A peptide of glycosyl-/glycerophosphate transferases involved in teichoic acid biosynthesis TagF/TagB/EpsJ/RodC was identified among the peptides obtained from *Enterococcus* strains.

Deacetylase enzymes were also among the proteins identified in *Enterococcus* spp. strains; these enzymes include a peptidoglycan-N-acetylglucosamine deacetylase, which plays a role in peptidoglycan deacetylation, avoiding eukaryotic lysozyme recognition during infection [101,102].

Finally, one peptide that exhibits weak similarity to O-antigen ligases was identified. This protein is involved in the synthesis of O-antigen, a side chain of the lipopolysaccharide found in the outer membrane in Gram-negative bacteria. Similar findings have been made in other Gram-positive bacteria, such as *Bacillus subtilis* [103].

### 2.7. Transporters Associated with Virulence Factors

Several ABC-type transporters are involved in virulence and play an important role in bacterial propagation during infection [104,105]. Seventy putative ABC transporters representing virulence factor peptides were identified in all *Enterococcus* strains, in addition to eight peptides corresponding to a variety of transporters that facilitate bacterial virulence strategies (Appendix A).

Bacteria, including pathogenic strains, are well-established to have a variety of strategies to survive harsh conditions, such as the conditions generated upon nutrient deprivation, responding in such cases with the release of several stress proteins as well as by immune evasion mechanisms. The LC-ESI-MS/MS analyses carried out here identified several peptides corresponding to proteins that are required for the uptake of metals, such as cobalt, copper and ferrochromium. Furthermore, some peptides identified as ABC transporters are involved in the transport of lantibiotic oligopeptides and peptides, amino acids, glycine/betaine [104], spermidine, putrescine, and multidrug transporters. One peptide of a lantibiotic ABC transporter was identified for the F13 strain, and three peptides of multidrug ABC transporters were identified for the F2 and F3 strains. Multidrug ABC efflux transporters extrude antibiotics out of the bacterial cells, thus allowing pathogenic bacteria to resist antimicrobial treatment [104,106].

Additionally, nonABC transporters related to virulence were also identified in the peptide analysis. These peptides belong to a variety of transporters, including the EamA/RhaT family transporter, cation diffusion facilitator family transporter, copper-transporting, MFS and drug efflux MFS transporters. The MFS (major facilitator superfamily) is one of the largest groups of solute transporters [107]. However, the cation diffusion facilitator family (CDF) is designed to resist increasing concentrations of divalent metal ions such as cadmium, zinc, and cobalt, facilitating their removal from the cells [108]. Moreover, many members of the EamA family proteins are classified as drug/metabolite transporters [109].

### 2.8. Other Bacterial Virulence Factors

Bacteriophages are well known to encode genes as bacterial virulence factors, including Panton-Valentine leucocidin, staphylokinase, enterotoxins, chemotaxis-inhibitory proteins and exfoliative toxins in *S. aureus* [24,110] and *Streptococcus* species [22,111,112]. These viruses are usually integrated into bacterial chromosomes as prophages, wherein they encode new properties in the host, or vice versa, as transcription may hardly be affected by gene disruptions [112]. Phage-encoded recombinases, rather than the host recombinase RecA, are involved in bacterial genome excisions and integrations [113,114]. In addition, bacteriophage and bacteria interactions may substantially alter the variability of the bacterial population [115,116].

Likewise, the presence of mobile genetic elements is considered a major mechanism to produce bacterial variability related to a possible mechanism(s) of horizontal plasmid transfer (HGT) among bacteria [117]. These mobile elements may be either plasmids or viral DNA fragments and provide a wide range of genes that encode proteins involved in antibiotic resistance, virulence determinants, and additional polypeptides playing important roles in a variety of metabolic pathways [23,118]. Many peptides corresponding to proteins involved in the acquisition of these mobile elements were identified in the current analysis, including recombinases and integrases. Additional peptides represented specific plasmid proteins, such as *E. faecalis* plasmid pPD1 bacI and transposases corresponding to different transposons.

Eleven of the characterized peptides (Appendix A) are indeed a part of different transposases. Among these peptides, several transposon insertion sequence (IS4, IS6, and IS30) peptides and three peptides of the transposase DDE superfamily proteins have been determined. These ISs have been associated with macrolide resistance genes in *Enterococcus* isolates [119]. Peptides belonging to recombinases and integrases, conjugative transposon protein, and one conjugal transfer protein, TraG, have been identified. TraG is an essential gene within the Tra operon for DNA transfer in bacterial conjugation [120].

Three peptides corresponding to a mutator family protein were also identified (Appendix A); this polypeptide belongs to the class II DNA transposable element (TE) family that can exchange ectopic genomic sequences, leading to the formation of new gene arrangements [121].

In addition, peptides that correspond to a PrgI family protein, a putative plasmid replication protein, and another corresponding to the *Enterococcus faecalis* plasmid pPD1 bacI were identified in this study, and collaborators observed that the bacteriocin-encoding plasmid pPD1 enhances the ability of *E. faecalis* to colonize the GIT. Strains harboring such a plasmid of pPD1 easily displace preexisting enterococcal strains lacking the plasmid, including vancomycin-resistant enterococci (VRE). Importantly, the pPD1-mediated colonization advantage required the resident bacteriocin synthesis operon [122]. Moreover, PrgI is encoded in plasmids of *E. faecalis*, but its function is still largely unknown [123]. A pheromone response system RNA-binding regulator PrgU was determined in this study. rgU is a protein of plasmid origin expressed mainly in *Enterococcus* bacteria. PrgU has been postulated to probably act as an RNA-binding regulator, mitigating the toxicity accompanying overproduction of PrgB-like adhesins, which are involved in conjugative transfer [124].

In addition, more proteins of phage origin and involved in genetic modifications were also identified, such as three peptides for clustered regularly interspersed short palindromic repeats (CRISPR)-associated endonuclease Cas proteins, a peptide of a YqaJ domain, and two peptides belonging to regulatory protein RecX, which might be regulators of RecA activity by interaction with the RecA protein or filament [125]. YqaJ forms part of the two-component SynExo viral recombinase functional unit [126].

A large number of phage peptides from structural proteins were identified (Appendix A). Peptides from proteins such as the major and minor capsid proteins, tail tape measure protein, phage portal protein, and phage tail fiber proteins were determined. The tape measure protein (TMP) determines the tail length and facilitates DNA entry into the cell during infection, and some TMPs have been reported to carry lysozyme-like and peptidase domains [127,128].

Several peptides of transposases, integrases, recombinases, and terminases were also identified. In this sense, a phage/plasmid primase P4 family domain protein and a PBSX family phage terminase peptide were determined, with the second involved in double-stranded DNA binding, DNA packaging, endonuclease, and ATPase activities [129].

Nine peptides from repressor-type Cro/CI were also determined. CI and Cro are encoded in the lysogeny module of lambdoid bacteriophages, particularly λ bacteriophages. Together, CII and CIII (which are formed through the antiterminator role of protein N) act as inducers that favor the first expression of the *cI* gene from the appropriate promoter; if the CI repressor predominates, the phage remains in the lysogenic state, but if the Cro predominates, the phage transitions into the lytic cycle, helped by the late Q regulator. The xenobiotic XRE regulator is extended in bacteria and has similarity to the Croλ repressor, exhibiting a helix-turn-helix (HTH) conformation [129]. Peptides of the CI/Cro-repressor types are usually named XRE family proteins in the National Center for Biotechnology Information (NCBI) database for bacteria.

Two peptides of a BppU family phage baseplate upper protein, several uncharacterized proteins that belong to bacteriophages and a phage infection protein YhgE were also found. BppU, also known as ORF48, is the N-terminal domain of baseplate upper proteins, which is found in bacteriophages [130]. Baseplate proteins are multiprotein molecular machines that control host cell recognition, attachment, tail sheath contraction, and viral DNA ejection [131].

Among phage peptides found in this analysis, some peptides identified when compared to the UniProtKB database were determined to be homologs to proteins found in different *Enterococcus* bacteriophages (Appendix A); other peptides were determined to be phage peptides found in *Enterococcus* bacteria. Proteins that we found of bacteriophage origin were either structural proteins, such as phage head protein gp7, phage tail sheath, tail length tape-measure protein, or altogether uncharacterized proteins (Appendix A).

## 3. Discussion

Due to the increased incidence of *Enterococcus* spp. in clinical microbiology and their presence in the food chain, the development of sensitive, rapid, and automated methods is required for the accurate identification and characterization of strains that can be implicated in food spoilage and food poisoning [132]. The LC-ESI-MS/MS proteome obtained from fourteen *Enterococcus* strains from dairy products identified 1403 nonredundant peptides. A total of 1327 peptides represent proteins that act as either virulence factors, toxins, or antibiotic resistance peptides, as well as tolerance proteins involved in bacterial resistance to toxic substances. Other identified proteins included colonization and immune evasion factors, polypeptides associated with antimicrobial production, ABC transporters, and other transporters related to virulence factors. The *Enterococcus* strains isolated from dairy products were previously characterized by MALDI-TOF-MS and phylogenetic analyses based on the 16S rRNA gene, and the genetic results were compared to the proteomic data. In addition, the analyses reported here involved an in-depth study of the antimicrobial and virulence factors present in the strains using shotgun proteomics tools. The rapid and accurate identification and potential pathogenicity characterization of pathogenic bacteria, including *Enterococcus* species, is an essential issue to maintain a good quality of the food chain.

As mentioned above, F5, F2, F1, F3, and F13 are enterococcal strains containing the highest number of virulence-related peptides identified (with 50, 44, 39, 37, and 32 peptides of virulence, respectively), indicating that these strains represent the most pathogenic bacteria compared with the other strains analyzed. The methodology described here allows determination of specific virulence factors as displayed by the bacteria in situ, as given in a particular environment and timing, hence allowing determination of the actual virulence status of the bacteria present in the food chain. This technology is a remarkable property. The technology may be used without previous growth of the bacterial strains that often do not show the same characteristics as the parental strains as they occur in the foodstuff.

All pathogen adaptations to either harsh environments or sublethal concentrations of antimicrobial agents contribute to the development of antimicrobial resistance, and, unfortunately, the last decades have seen a constant increase in antibiotic resistance phenotypes in enterococcal strains thus far isolated from foodstuffs [133,134].

The *Enterococcus* strains studied here contain many peptides involved in their resistome, including resistance to penicillin that is mediated by penicillin-binding proteins and other antibiotics, such as the MarR family of transcriptional regulators, TetR proteins, the PASTA domain, and the GNAT family of acetyltransferases. A variety of different drug transport pathways were also found, such as the multidrug efflux MFS transporter.

There is no doubt that, today, it is even more essential than ever that pathogenic bacterial strains be quickly identified both in foodstuffs and in clinics to provide the appropriate antimicrobial treatment. The present report shows that bacterial characterization can be quickly achieved by the use of LC-ESI-MS/MS. In fact, there is an urgent need for novel therapies for both the treatment and prevention of diseases caused by pathogenic species of enterococci. Bacteriocins are active against antibiotic-resistant bacterial strains, and, accordingly, they can be used, either independently or in combination with other antimicrobials, to cope with these infections. Indeed, bacteriocin and related peptides have been found here, and a deeper analysis should be performed to ascertain their effectiveness as antimicrobials.

Some reports have demonstrated how strains isolated from food and/or food production environments that harbor plasmids and bacteriophages in their genome provide important advantages for survival in food or associated environments, and, notably, the presence of mobile genetic elements is indeed considered a major mechanism of antibiotic resistance acquisition [6,7].

Finally, many peptides corresponding to proteins that play a role in the colonization and immune evasion of pathogenic enterococcal strains were identified in this study. These proteins play a crucial role in bacterial internalization into mammalian cells during the course of infection; thus, the identification of microbial peptides may provide a positive way to characterize the pathogen [4].

The precise proteomic method implemented in the present study represents a useful step for further analyses of pathogenic bacteria, as it offers considerable advantages over traditional approaches in terms of speed and reliability, without the need for full genomic sequencing and analysis or pregrowth of the bacterial strains [19,21,22,24,135].

## 4. Materials and Methods

### 4.1. Bacterial Strains

Fourteen *Enterococcus* strains isolated from dairy products were used in this study (Table 1). Strains were previously characterized by MALDI-TOF-MS and 16S rRNA sequencing [132]. The *Enterococcus* strains were grown in brain heart infusion (BHI, Oxoid Ltd., Hampshire, UK) at 31 °C for 24 h. Bacterial cultures were then transferred to plate count agar (PCA, Oxoid Ltd., Hampshire, UK) and subjected to further incubation at 31 °C for 24 h.

### 4.2. Protein Extraction

Protein extraction was performed as indicated previously [19]. In short, a fresh inoculation loop of bacterial culture was resuspended in 100 μL of a solution with 50% acetonitrile (ACN; Merck, Darmstadt, Germany) and 1% aqueous trifluoroacetic acid (TFA; Acros Organics, Bridgewater, NJ, USA). After vortexing and centrifuging, the supernatant was further treated with lysis buffer consisting of 60 mM Tris-HCl pH 7.5, 1% lauryl maltoside, 5 mM phenylmethanesulfonyl fluoride (PMSF), and 1% dithiothreitol (DTT). The supernatant was transferred to a new tube, and the amount of protein was determined by the bicinchoninic acid method (Sigma Chemical Co., St. Louis, MO, USA). Due to the difficulties to achieve the main 14 different Enterococcus strains that affect to dairy products, *Enterococcus faecalis* (9 different strains), *Enterococcus durans* (1 strain), and *Enterococcus faeceium* (4 different strains), only one biological replicate for each strain was used in the manuscript.

### 4.3. Peptide Sample Preparation

Protein extracts were solubilized and further digested with trypsin, as reported previously [136]. To do so, 100 μg of protein was dried in a SpeedVac (CentriVap, Labconco Co., Kansas City, MO, USA), resuspended in 25 μL of denaturation buffer (8 M urea in 25 mM ammonium bicarbonate, pH 8.0), and sonicated for 5 min. Then, the addition of DTT followed, at a final concentration of 10 mM, and incubation at 37 °C for 1 h. Alkylation was easily achieved by addition of the appropriate amount of iodoacetamide (IAA) to a final concentration of 50 mM; the solution was dark-incubated for 1 additional hour at room temperature. The sample was diluted with 4 volumes of 25 mM ammonium bicarbonate (pH 8.0) to reduce the urea concentration. The final step included trypsin digestion (Promega, Madison, WI, USA) with a protease:protein ratio of 1:100. The incubation was performed overnight at 37 °C.

### 4.4. Shotgun LC–ESI–MS/MS Analysis

The peptide digests prepared as shown before were acidified with formic acid (FA) (~pH 2), desalted in a C18 MicroSpin™ column (The Nest Group, Southborough, MA, USA), and finally analyzed by LC-ESI-MS/MS using a Proxeon EASY-nLC II Nanoflow system (Thermo Fisher Scientific, San Jose, CA, USA) coupled to an LTQ-Orbitrap XL mass spectrometer (Thermo Fisher Scientific) [19,137]. Peptide separation (2 μg) was performed in a reverse-phase (RP) column (EASY-Spray column, 50 cm × 75 μm ID, PepMap C18, 2 μm particles, 100 Å pore size, Thermo Fisher Scientific) equipped with a 10 mm precolumn (Accucore XL C18, Thermo Fisher Scientific). Elution from the column was performed by means of a linear gradient from 5 to 35% solvent B (solvent A: 98% water, 2% ACN, 0.1% FA; solvent B: 98% ACN, 2% water, 0.1% fatty acid (FA)) for 120 min at a flow rate of 300 nL/min. Electrospray ionization was carried out with a spray voltage of 1.95 kV at a capillary temperature of 230 °C. Peptides were analyzed in positive mode (1 μscan; 400 to 1600 amu), followed by 10 data-dependent collision-induced dissociation (CID) MS/MS scans (1 μscan), using an isolation width of 3 amu and a normalized collision energy of 35%. After the second fragmentation event, dynamic exclusion was set for 30 s, and ions with an unassigned charge state were excluded from MS/MS analysis.

### 4.5. LC-ESI-MS/MS Data Processing

The MS/MS spectra obtained by LC-ESI-MS/MS were analyzed using the program SEQUEST-HT (Proteome Discoverer 2.4, Thermo Fisher Scientific) and compared to the *Enterococcus* UniProt/TrEMBL protein database (containing 431,881 protein sequence entries, September 2020). MS/MS spectra were searched using fully tryptic cleavage constraints, and up to two missed cleavage sites were allowed. Tolerance windows were set at 10 ppm for precursor ions and 0.06 Da for MS/MS fragment ions. The variable modifications allowed were (M*) methionine oxidation (+15.99 Da) and protein N-terminal acetylation (+42.0106 Da). Carbamidomethylation of cysteine (Cys) (+57.02 Da) (C*) was considered a fixed modification. The percolator algorithm (Käll et al., 2007) was used to validate the results as well as for statistical analysis. The peptide false discovery rate (FDR) was always kept at less than 1%. The mass spectrometry proteomics data have been deposited to the ProteomeXchange Consortium via the PRIDE [26] partner repository with the dataset identifier PXD036435.

Label-free quantification (LFQ) of relative protein abundance for each strain were performed using the Proteome Discover 2.4 program (Thermo Fisher Scientific), using the Minora Feature Detector node and the ANOVA (individual proteins) approach. Peak areas of ion features corresponding to the same peptide in different charge forms were summed up to one value.

## Figures and Tables

**Figure 1 ijms-23-10971-f001:**
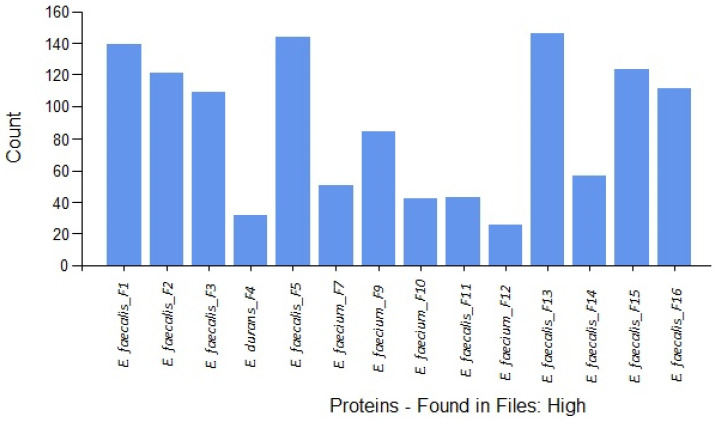
Distribution of the high-abundance proteins for each Enteroccocus strain determined by LFQ.

**Figure 2 ijms-23-10971-f002:**
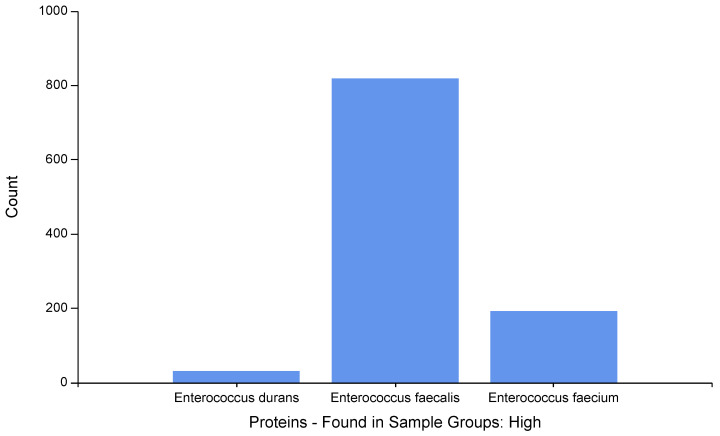
Distribution of the high-abundance proteins for each Enteroccocus species determined by LFQ.

**Figure 3 ijms-23-10971-f003:**
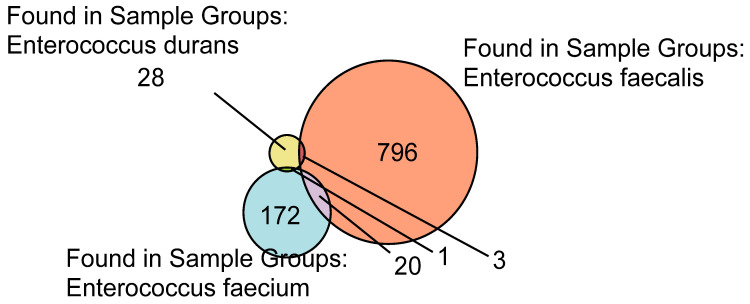
Venn diagram of the high-abundance proteins for all the Enteroccocus samples determined by LFQ.

**Table 1 ijms-23-10971-t001:** *Enterococcus* strains used in this study. Total virulence factor peptides represent the number of peptides identified by LC–ESI-MS/MS. CECT—Spanish Type Culture Collection. F3 lacks a GenBank accession number because the sequencing chromatogram did not show enough quality.

Sample	Species	Strain	Source	GenBank Accession Number	Total Virulence Factor Peptides
F1	*Enterococcus faecalis*	ISPA118	Milk	OP113834	39
F2	*Enterococcus faecalis*	ISPA 10	Dairy product	KC510240	44
F3	*Enterococcus faecalis*	ISPA T07	Milk	- *	37
F4	*Enterococcus durans*	CECT 411	Milk powder	KC510230	5
F5	*Enterococcus faecalis*	ISPA 28	Cheese	OP113835	50
F7	*Enterococcus faecium*	ISPA FRP2	Cheese	KC510246	14
F9	*Enterococcus faecium*	ISPA ID114	Milk	KC510249	19
F10	*Enterococcus faecium*	ISPA G07	Cheese	KC510247	4
F11	*Enterococcus faecalis*	ISPA ID119	Dairy product	KC510243	7
F12	*Enterococcus faecium*	ISPA G010	Cheese	KC510248	2
F13	*Enterococcus faecalis*	MORA 13	Dairy product	KC510244	32
F14	*Enterococcus faecalis*	ISPA ID 116	Milk	OP113836	12
F15	*Enterococcus faecalis*	ISPA 148	Dairy product	KC510245	25
F16	*Enterococcus faecalis*	ISPA 89	Dairy product	KC510242	20
Total peptides					310

* F3 lacks a GenBank accession number because the sequencing chromatogram was not of a high enough quality.

**Table 2 ijms-23-10971-t002:** Proteins corresponding to bacterial resistance to antibiotics, antimicrobial-related proteins, and other virulence factors, identified in the *Enterococcus* strains analyzed.

Function	Protein
Antibiotic resistance	TetR family transcriptional regulator
	TetR family transcriptional regulator
	N-acetyltransferase domain-containing protein
	PadR domain-containing protein
	GNAT family acetyltransferase
	Cass2 domain-containing protein
	MarR family transcriptional regulator
	Penicillin-binding protein
	Daunorubicin resistance protein DrrC
	OmpR/PhoB-type domain-containing protein
	PASTA domain-containing protein
	VanY D-Ala-D-Ala carboxypeptidase
Additional resistances and tolerances	Cold-shock protein
	General stress protein
	CsbD-like protein
	YitT family protein
	Tellurite resistance protein TelA
	MerR family transcriptional regulator
	SugE protein
	Chaperone protein DnaK
Antimicrobial compounds	Lantibiotic biosynthesis protein
	Type 2 lantipeptide synthetase LanM
	LanM family lanthionine synthetase
	Thiopeptide-type bacteriocin biosynthesis domain protein
	Bacteriocin Protein
	Radical SAM additional 4Fe4S-binding SPASM domain-containing protein
Toxin	Type II toxin–antitoxin system PemK/MazF family toxin
	Type II toxin–antitoxin system RelE/ParE family toxin
	Toxin–antitoxin system, antitoxin component, AbrB family
	Toxin PIN
	Exfoliative toxin A/B
	Prevent-host-death family antitoxin (Phd antitoxin)
	LXG domain-containing protein
Colonization and immune evasion factors	Internalin
	Sortase
	LPXTG cell wall anchor domain-containing protein
	Adhesin BspA
	Fibronectin/fibrinogen-binding protein
	Collagen-binding protein
	Ig domain-containing protein
	DUF4097 domain-containing protein
	Flagellar hook-associated protein 2
	Fn3-like domain-containing protein
	Fimbrial isopeptide formation D2 domain-containing protein
	SpaA domain-containing protein
	Endopeptidase NlpC/P60 family protein
	M24/M78/M28/M20/M25/M40 family peptidases
	Peptidase T
	DD-transpeptidase
	Dipeptidase PepV
	Endopeptidase La
	S8/S9/S74 Family Peptidases
	Dipeptidyl aminopeptidase
	Peptidase U32
	Peptidase S74
	Proline dipeptidase
	Oligoendopeptidase PepF/M3 family protein
	Peptidase C51
	ImmA/IrrE family metallo-endopeptidase
	Isoaspartyl dipeptidase
	Signal peptidase I
	Immune inhibitor A
	ClpA protease
	ClpC protease
	Zinc protease
	Capsular polysaccharide biosynthesis protein CpsC
	N-acetylmuramoyl-L-alanine amidase
	LysM domain protein
	Hemolysin III family channel protein
	Autolysin modifier protein
	Mga domain-containing protein
	Toxin secretion/phage lysis holin
	Type VII secretion protein EssC
	LysR family transcriptional regulator
	LytTR family transcriptional regulator
	HTH-type transcriptional regulator KdgR
	ArpU family transcriptional regulator
	Competence protein ComEA helix-hairpin-helix repeat region
	Control of competence regulator ComK, YlbF/YmcA
	Regulatory protein YlbF
	Spore coat protein
	Sporulation protein YjcZ
	Restriction endonuclease type IV, Mrr
	SfeI restriction endonuclease
	Type-2 restriction enzyme
	O-antigen ligase
	Methyl-accepting chemotaxis protein (MCP) signaling domain
	N-acetylglucosamine-6-phosphate deacetylase
	Glycosyl/glycerophosphate transferases involved in teichoic acid biosynthesis TagF/TagB/EpsJ/RodC
ABC transporters	Copper ABC transporter permease
	Ferrichrome ABC transporter FhuC
	Cobalt ABC transporter permease
	Lantibiotic protection ABC transporter
	Multidrug ABC transporter
	Spermidine/putrescine ABC transporter
	Glycine betaine ABC transporter
	Peptide ABC transporter
	Amino acid ABC transporter
	Excinuclease ABC subunit A
	Sulfate ABC transporter ATP-binding protein
	C4-dicarboxylate ABC transporter
	Multiple sugar ABC transporter substrate-binding protein
	Sugar ABC transporter
	Carbohydrate ABC transporter substrate-binding protein
	Heme ABC transporter
	Thiol reductant ABC exporter subunit CydC
Other transporters	Major facilitator superfamily (MFS) transporter
	Multidrug resistance MFS transporter
	Cation diffusion facilitator family transporter
	EamA/RhaT family transporter
	Copper-transporting ATPase CopB
Alternative virulence factors	Transposase
	Tnp-DDE superfamily
	IS4 family transposase
	Transposase InsI for insertion sequence element IS30C
	IS30/IS4/IS6 family transposaseS
	Conjugative transposon protein
	Conjugal transfer protein TraG
	Mutator family transposase
	Tyrosine-type recombinase/integrase
	*Enterococcus faecalis* plasmid pPD1 bacI
	Putative plasmid replication protein
	PrgI family protein
	Pheromone response system RNA-binding regulator PrgU
	Regulatory protein RecX
	CRISPR-associated endonuclease Cas9
	CRISPR-associated endonuclease Cas10
	CRISPR-associated endoribonuclease Cas2
	YqaJ domain-containing protein
	Luciferase family oxidoreductase, group 1
Phage proteins	Phage capsid proteins
	Phage tail proteins
	Phage portal protein
	Phage/plasmid primase, P4 family domain protein
	PBSX family phage terminase
	Phage terminase
	Phage integraseç
	Cro/CI family transcriptional regulator
	BppU family phage baseplate upper protein
	phage infection protei YhgE

**Table 3 ijms-23-10971-t003:** Total number of peptides corresponding to virulence factors organized in groups according to the main role they play, identified in the *Enterococcus* strains analyzed.

Function	Total Virulence Factor Peptides
Colonization and immune evasion factors	104
ABC transporters	70
Other transporters	8
Alternative virulence factors	32
Phage proteins	49
Antibiotic resistance	21
Additional resistances and tolerances	11
Antimicrobial compounds	6
Toxins	9
Total Peptides	310

## Data Availability

Not applicable.

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
