# Peer review of "Proteomic Characterization of Virulence Factors and Related Proteins in Enterococcus Strains from Dairy and Fermented Food Products"

_ijms, 2022, doi:10.3390/ijms231810971_

Round 1
Reviewer 1 Report
In this study, the authors examined the Enterococcus species that have been identified as the causal agents of health care-associated illnesses in dairy and fermented food items. To examine the proteomics signature, the authors identified a total of 1403 nonredundant peptides, representing 1327 proteins, using mass spectrometry as the primary approach. Moreover, 310 of these peptides related to proteins that have a direct function as virulence factors 29 in the pathogenicity of Enterococcus. Therefore, the authors concluded that the acquired global proteome library may serve as a foundation for future research on pathogenic Enterococcus species, therefore helping the development of new therapies. There is huge drawback in the technical analysis and reporting of the data.
In the proteomics study what is the number of replicates were used for each data is not represented.
It is important to describe why the depth of the proteome data is soo low.
Where the proteomics data is submitted is also not reported.
Why not to perform the label free quantification of all the organism used. Followed with making the enrichment analysis. This would be the best of representing this kind of data.
Other way of detailing this proteomics data could be to show the abundance of each protein and respective peptide iBAQ or emPAI or similar calculation. It is based on the choice of the authors but in current form the authors have not utilised the importance of proteomics data sets.
Minor comment
The introduction is too long and loses attention at some point. Rather it would be beneficial for the article to keep the focus and not report too much of the background which is not relevant to the study.
Author Response
In this study, the authors examined the Enterococcus species that have been identified as the causal agents of health care-associated illnesses in dairy and fermented food items. To examine the proteomics signature, the authors identified a total of 1403 nonredundant peptides, representing 1327 proteins, using mass spectrometry as the primary approach. Moreover, 310 of these peptides related to proteins that have a direct function as virulence factors 29 in the pathogenicity of Enterococcus. Therefore, the authors concluded that the acquired global proteome library may serve as a foundation for future research on pathogenic Enterococcus species, therefore helping the development of new therapies. There is huge drawback in the technical analysis and reporting of the data.
In the proteomics study what is the number of replicates were used for each data is not represented.
Thank you very much for your opinion. In order to improve the manuscript, your following suggestions have been taken into consideration and answered to the best of our knowledge with detailed proof-reading.
- Due to the difficulties to achieve the main 14 different Enterococcus strains that affects to dairy products: Enterococcus faecalis (9 different strains), Enterococcus durans (1 strain) and Enterococcus faeceium (4 different strains) only one biological replicate for each strain were used in the manuscript. According to the referee suggestion we indicated that in the Materials and Methods section.
It is important to describe why the depth of the proteome data is soo low.
According to the referee suggestion we indicated in the Results section that although some intracellular proteins are present, the depth of the proteome results corresponded to surface-associated proteins mainly after the protein extraction preparation methods described in Materials and Methods. This is due that the majority of virulence factors are localized in the surface of bacteria, either secretory and membrane associated (Chajecka-Wierzchowska et al., LWT 2017; Sharma AK, Ind. J Microbiol. 2017) and we prepared the protein extraction to increase its identification. Additionally, because we validated the results with a high restriction for a FDR < 1%.
Where the proteomics data is submitted is also not reported.
According to the referee suggestion proteomics data are in Suplementary Data 2 and 3 of the manuscript and now were submitted to the public ProteomeXchange repository with the ID PXD036435. This code was also included in the results section.
Why not to perform the label free quantification of all the organism used. Followed with making the enrichment analysis. This would be the best of representing this kind of data.
According to the referee suggestion we did Label Free Quantification (LFQ) of all strains used in the study by Proteome Discoverer 2.4 program using Minora Feature Detection for precursor ion quantifier and ANOVA for individual proteins to identify the ratios based on protein abundances We included the details of the procedure in Materials and Methods section 4.5 and the results in Supplementary Data 3.
Other way of detailing this proteomics data could be to show the abundance of each protein and respective peptide iBAQ or emPAI or similar calculation. It is based on the choice of the authors but in current form the authors have not utilised the importance of proteomics data sets.
According to the referee suggestion we determined the abundance of each protein using a similar calculation based on Label Free Quantification (LFQ) by Minora Feature Detection and ANOVA that is included in the Proteome Discoverer 2.4 program. We included the details of the procedure in Materials and Methods (section 4.5) and the data in Supplementary Data 3. In addition, a new Results section (2.2. Label-Free Quantification (LFQ) of Enterococcus species) and new Figures 1, 2 and 3 were included in the manuscript.
Minor comment
The introduction is too long and loses attention at some point. Rather it would be beneficial for the article to keep the focus and not report too much of the background which is not relevant to the study.
According to the referee suggestion we have rewrote the introduction.

Reviewer 2 Report
Enterococcus spp. are a widespread group of Gram-positive bacteria. Normal gastrointestinal tract inhabitants, they play a beneficial role in the dairy and meat industry, especially during the ripening of traditional cheeses and sausages. However, some Enterococcus spp. such as Enterococcus faecalis and E. faecium are opportunistic pathogens. They readily form biofilms, are often resistant to antibiotics, and can cause a wide range of infections in human infants and adults. The presence of chromosomally integrated bacteriophages and plasmids in enterococcal genomes contributes to their variability and virulence-related advantages such as attachment to host cells and immune evasion. Development of effective therapy for infections caused by these organisms requires identification of the enterococcal strain involved as well as its virulence factors and antibiotic sensitivities. The present study describes the use of liquid chromatography-electrospray ionization tandem mass spectrometry (LC-ESI-MS/MS) to perform a global shotgun proteomic characterization for opportunistic pathogenic enterococci from different dairy and fermented food products. This method identified a total of 1,403 nonredundant peptides, representing 1,327 proteins, with 310 of these peptides corresponding to proteins playing a direct role as virulence factors for Enterococcus pathogenicity. The global proteomic repository obtained here can be the basis for further research into pathogenic Enterococcus spp. and development of novel therapeutics. They also provide a broad context for epidemiological investigations of these bacteria from a One Health approach. (Adapted from the Abstract)
Enterococcus spp. play a major beneficial role in the dairy and meat industries, but they also are opportunistic pathogens of humans both on their own and as part of polymicrobial infections. Studies of the mechanisms and virulence factors involved in enterococcal colonization and pathogenesis have been somewhat limited in the past. The authors have conducted a proteomic study of 14 strains of E. faecalis, E. durans, and E. faecium isolated from various dairy sources using LC-ESI-MS/MS to identify proteins involved in bacterial resistance to antibiotics, bacterial toxicity, host colonization and immune evasion, enterococcal virulence factors and proteins associated with these factors. The findings in the body of the text and in the supplementary files are clearly presented and well discussed. They thus represent an important advance in this area of proteomics.
There are some easily remedied difficulties.
1. The text is clearly written and easily understandable. Unfortunately, the Abstract is considerably less focused, and would benefit from careful revision to emphasize the critically important information application of LC-ESI-MS/MS enabled the authors to obtain.
2. Table 1. The source of samples F2, F11, F13, F15 and F16 is given as “Dairy”. What is meant by this? Were the bacteria isolated from the barn in which the animals were housed? From both milk and cheese? From machinery used in preparation of dairy products? This should be be better specified or explained in a footnote to the Table.
3. Table 1. Why does sample F3 lack a GenBank accession number? It should be submitted and one obtained. If for some reason this is not possible, this should be explained in a footnote to the Table.
Author Response
Thank you very much for your opinion. In order to improve the manuscript, your following suggestions have been taken into consideration and answered to the best of our knowledge with detailed proof-reading.
There are some easily remedied difficulties.
1. The text is clearly written and easily understandable. Unfortunately, the Abstract is considerably less focused, and would benefit from careful revision to emphasize the critically important information application of LC-ESI-MS/MS enabled the authors to obtain.
According to the referee suggestion we have rewritten the abstract.
- Table 1. The source of samples F2, F11, F13, F15 and F16 is given as “Dairy”. What is meant by this? Were the bacteria isolated from the barn in which the animals were housed? From both milk and cheese? From machinery used in preparation of dairy products? This should be be better specified or explained in a footnote to the Table.
In order to provide a better understanding of the term, we have changed “dairy” to “dairy product”. We don’t have more information for these strains that they were obtained from a milk-derived product.
- Table 1. Why does sample F3 lack a GenBank accession number? It should be submitted and one obtained. If for some reason this is not possible, this should be explained in a footnote to the Table.
According to the referee suggestion we indicated in Table 1 that F3 lacks a GenBank accession number because the sequencing chromatogram was not quality enough.

Round 2
Reviewer 1 Report
No more questions